# LAPAR: Linearly-Assembled Pixel-Adaptive Regression Network for Single Image Super-Resolution and Beyond

**Wenbo Li**[1]* **Kun Zhou**[2]* **Lu Qi**[1] **Nianjuan Jiang**[2] **Jiangbo Lu**[2]† **Jiaya Jia**[1,2]

[1]The Chinese University of Hong Kong  [2]Smartmore Technology

{wenboli,luqi,leojia}@cse.cuhk.edu.hk
{kun.zhou,nianjuan.jiang,jiangbo}@smartmore.com

## Abstract

Single image super-resolution (SISR) deals with a fundamental problem of up-sampling a low-resolution (LR) image to its high-resolution (HR) version. Last few years have witnessed impressive progress propelled by deep learning methods. However, one critical challenge faced by existing methods is to strike a sweet spot of deep model complexity and resulting SISR quality. This paper addresses this pain point by proposing a linearly-assembled pixel-adaptive regression network (LAPAR), which casts the direct LR to HR mapping learning into a linear coefficient regression task over a dictionary of multiple predefined filter bases. Such a parametric representation renders our model highly lightweight and easy to optimize while achieving state-of-the-art results on SISR benchmarks. Moreover, based on the same idea, LAPAR is extended to tackle other restoration tasks, e.g., image denoising and JPEG image deblocking, and again, yields strong performance.

## 1 Introduction

Single image super-resolution aims at reconstructing a high-resolution (HR) image from a low-resolution (LR) one. Due to its wide applications, intensive research efforts and great progress have been made in the past decades. Among existing methods, the simplest one is to adopt basic spatially invariant nearest-neighbor, bilinear and bicubic interpolation. However, these simple linear methods neglect the content variety of natural images, and usually overly smooth the structures and details.

To better adapt to different image elements, sparse dictionary learning processes pixels (or patches) individually. Early work [1, 2, 3, 4] learned a pair of dictionaries of LR and HR patches where sparse coding is shared in the LR and HR space. During inference, given trained dictionaries, only sparse coding is optimized to complete estimation. Dictionary-based methods generally yield stable results. However, the difficulty of joint optimization in the training process still affects recovery performance. Taking an example-based learning approach, Romano *et al.* [5] hashed image patches into clusters based on local gradient statistics, and a single filter is constructed and applied per cluster. Despite simple, such a hard-selection operation is discontinuous and non-differential. Meanwhile, it only provides compromised rather than optimal solutions for varying input patterns.

Recently, a great number of deep learning methods [6, 7, 8, 9, 10, 11, 12, 13, 14, 15, 16] were proposed to predict the LR-HR mapping. The challenge lies in the unconstrained nature of image contents [17], where training can be unstable when largely varying stochastic gradients exist. It causes annoying artifacts. Residual learning, attention mechanism and other strategies were used to alleviate these issues. Also, this line of methods is highly demanding on computational resources.

---

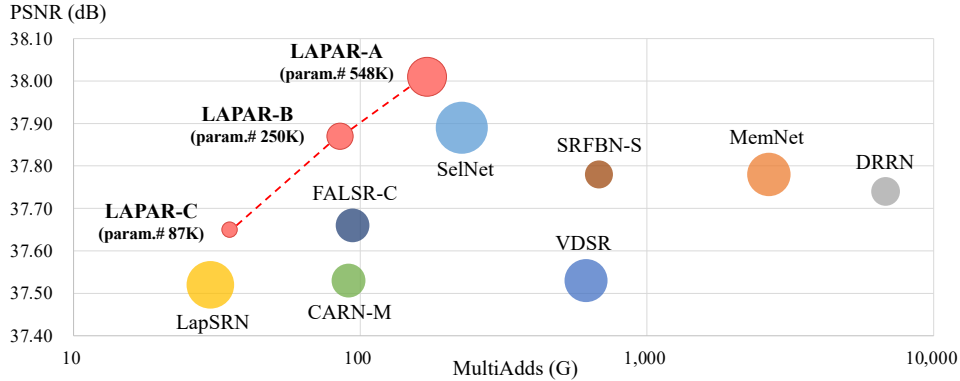

Figure 1: Comparison between our proposed LAPAR and other lightweight methods ($< 1$M parameters) on Set5 [27] for $\times 2$ setting. Circle sizes are set proportional to the numbers of parameters.

Different from the straightforward estimation, adaptive filters (kernels) can group neighboring pixels in a spatially variant way. It has been proven effective in tasks of super-resolution [18, 19, 20], denoising [17, 21], video interpolation [22] and video prediction [23, 24, 25]. The benefits brought in are twofold. First, the estimated pixels always lie within the convex hull of surroundings to avoid visual artifacts. Second, the network only needs to evaluate the relative importance of neighbors rather than predicting absolute values, which speeds up the learning process [26, 17]. Nevertheless, many methods estimate filters without regularization terms, which are useful to constrain the solution space of the ill-posed SISR problem.

In this paper, we tackle the efficient learning and reconstruction problem of SISR by utilizing a linear space constraint, and propose a linearly-assembled pixel-adaptive regression network (LAPAR). The core idea is to regress and apply pixel-adaptive "enhancement" filters to a cheaply interpolated image (by bicubic upsampling). Specifically, we design a lightweight convolutional neural network to learn linear combination coefficients of predefined filter bases for every input pixel. Despite spatially variant, the estimated filters always lie within a linear space assembled from atomic filters in our dictionary, and are easy and fast to optimize. In combination with the LAPAR network to achieve state-of-the-art SISR results, our dictionary of filter bases can be surprisingly common and simple, which are basically dozens of (anisotropic) Gaussian and difference of Gaussians (DoG) kernels.

The overall contributions of this paper are threefold:

- We propose LAPAR for SISR. Figure 1 shows that LAPAR achieves state-of-the-art results with the least model parameters and MultiAdds among all existing lightweight networks.

- Different from previous methods, we predefine a set of meaningful filter bases and turn to optimize assembly coefficients in a pixel-wise manner. Extensive experiments demonstrate the advantages of this learning strategy, as well as its strength in accuracy and scalability.

- Based on the same framework, LAPAR can also be easily tailored to other image restoration tasks, e.g., image denoising and JPEG image deblocking, and yields strong performance.

## 2 Method

Before presenting the LAPAR approach, we first give a brief revisit to image super-resolution.

### 2.1 Revisiting Image Super-Resolution

In the single image super-resolution (SISR) task, for a high-resolution (HR) vectorized image $\mathbf{y} \in \mathbb{R}^{HWs^2}$, the low-resolution (LR) counterpart $\mathbf{x} \in \mathbb{R}^{HW}$ is generally formulated as

$$\mathbf{x} = \mathbf{SHy}, \tag{1}$$

where $\mathbf{H} \in \mathbb{R}^{HWs^2 \times HWs^2}$ is a blurring filter and $\mathbf{S} \in \mathbb{R}^{HW \times HWs^2}$ represents the downsampling operator. The goal of SISR is to recover $\mathbf{y}$ from the given LR image $\mathbf{x}$. However, it is clear that

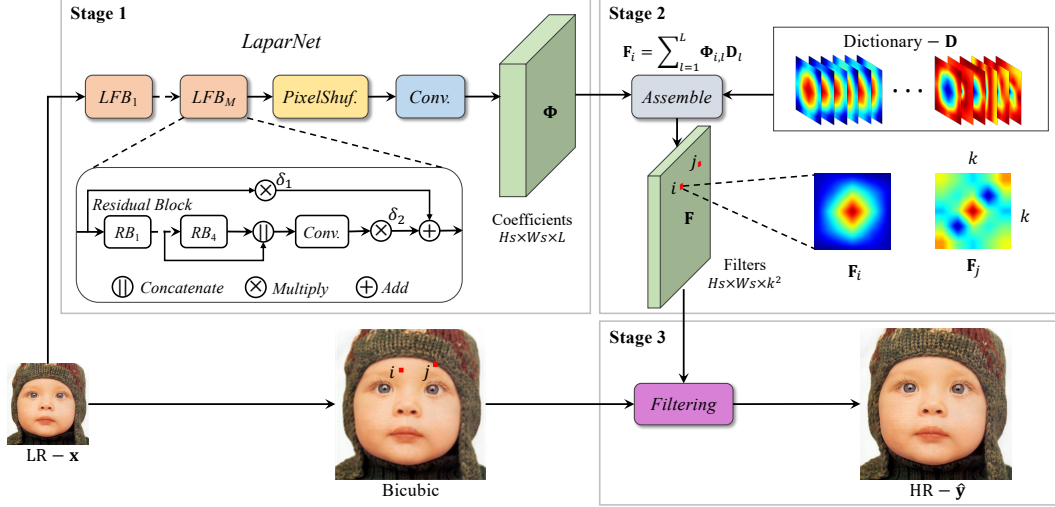

Figure 2: Framework of linearly-assembled pixel-adaptive regression network (LAPAR). Our method consists of three primary stages, i.e., stage 1: regressing linear combination coefficients; stage 2: assembling pixel-adaptive filters; stage 3: applying adaptive filters to the bicubic upsampled image.

SISR is an ill-posed problem since infinite possible solutions of $\mathbf{y}$ can be derived. Apart from the reconstruction constraints, more image priors or constraints are required to tackle this problem.

## 2.2 Our Learning Strategy

Instead of estimating the LR-HR mapping, we learn the correspondence between the cheaply interpolated (e.g., by bicubic upsampling) image and HR to construct our fast and robust solution.

In the first step, we extend the cheaply upsampled result to $\mathbf{B} \in \mathbb{R}^{HWs^2 \times k^2}$, a matrix consisting of $HWs^2$ patches with size $k^2$ ($k = 5$ in this paper). The $i$-th target pixel $\mathbf{y}_i$ is predicted by integrating neighboring pixels $\mathbf{B}_i$ (the $i$-th row of $\mathbf{B}$) centered at the coordinate of $\mathbf{y}_i$. The objective function is formulated as

$$\min_{\mathbf{F}_i} \left\| \mathbf{F}_i \mathbf{B}_i^T - \mathbf{y}_i \right\|_2^2, \tag{2}$$

where $\mathbf{F} \in \mathbb{R}^{HWs^2 \times k^2}$ is the filter matrix that needs to be estimated. In our method, filters are learned in a spatially variant way for rich visual details.

To deal with ill-posed problems, previous works have adopted different regularization terms, such as $l_1$-norm [1, 2, 3, 4], $l_2$-norm [28], total variation (TV) [29, 30] and anisotropic diffusion [31]. With the regularization term, the optimization objective of $\mathbf{F}_i$ is defined as follows,

$$\min_{\mathbf{F}_i} \left\| \mathbf{F}_i \mathbf{B}_i^T - \mathbf{y}_i \right\|_2^2 + \lambda R \left( \mathbf{F}_i \right), \tag{3}$$

where $\lambda$ is a balancing parameter and $R$ is the regularization function. It is nontrivial to determine unifying regularization that can be optimized efficiently and is also general enough for different tasks. Thus, instead of designing a hand-crafted term, we introduce a linear space constraint to facilitate the optimization, which is analogous to the idea of the popular parametric human body space (e.g., SMPL [32]) in spirit. Specifically, a filter $\mathbf{F}_i$ in our method is represented as a linear combination of a set of underlying base filters:

$$\mathbf{F}_i = \mathbf{\Phi}_i \mathbf{D}, \tag{4}$$

where $\mathbf{D} \in \mathbb{R}^{L \times k^2}$ represents a dictionary consisting of $L$ filter bases, and $\mathbf{\Phi} \in \mathbb{R}^{HWs^2 \times L}$ refers to the linear combination coefficients. Unlike traditional dictionary learning methods [1, 2, 3, 4] that optimize $\mathbf{D}$ and $\mathbf{\Phi}$ simultaneously, we simplify the learning procedure and only optimize coefficients $\mathbf{\Phi}$ by predefining a meaningful dictionary as detailed in Sect. 2.3. It is clear that the proposed strategy is well-conditioned as long as the dictionary $\mathbf{D}$ is over-complete and can well represent the desired $k^2$-sized filters for different pixels. Now, optimization of $\mathbf{F}_i$ becomes optimizing $\mathbf{\Phi}_i$ equivalently:

$$\min_{\mathbf{\Phi}_i} \left\| \mathbf{\Phi}_i \mathbf{D} \mathbf{B}_i^T - \mathbf{y}_i \right\|_2^2. \tag{5}$$

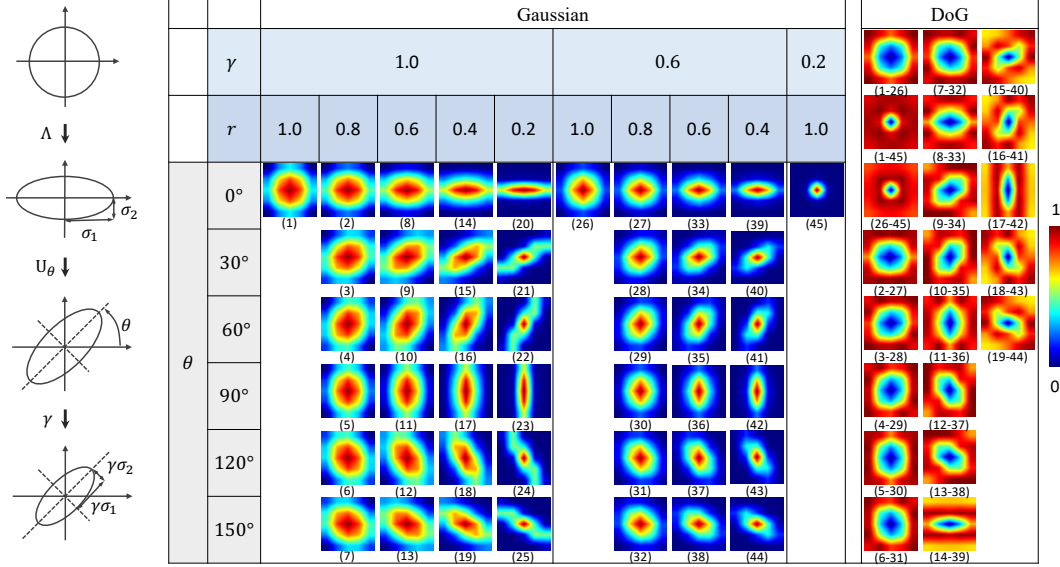

Figure 3: Visualization of part of the filters in the proposed dictionary. Symbols $\gamma, \theta, r$ describe scaling, rotation and elongation ratio ($r = \sigma_2/\sigma_1$). The Gaussian is denoted by (a), and DoG (a-b) means the difference of Gaussian (a) and Gaussian (b). When $\gamma = 1$ and $r = 0.2$, there is a Gaussian every $15°$ (six of them omitted from display). For better visualization, filters are normalized to $[0, 1]$.

We propose a convolutional network (i.e., $LaparNet$) as shown in Figure 2 to estimate the coefficient matrix $\mathbf{\Phi} = LaparNet(\mathbf{x})$. The network details will be presented in Sect. 2.4. Based on the regressed linear coefficients $\mathbf{\Phi}_i$ for the $i$-th pixel, its high-resolution prediction $\hat{\mathbf{y}}_i$ is derived as

$$\hat{\mathbf{y}}_i = \mathbf{\Phi}_i \mathbf{DB}_i^T . \tag{6}$$

In the training process, we use the Charbonnier loss as the error metric, which takes the form of

$$Loss = \sqrt{\|\hat{\mathbf{y}} - \mathbf{y}\|_2^2 + \epsilon^2} , \tag{7}$$

where $\epsilon$ is a small constant. The parameters of the network are optimized by gradient descent. Without the bells and whistles, our proposed method obtains decent performance as illustrated in Sect. 3.

## 2.3 Dictionary Design

In this paper, we provide a redundant dictionary with 72 underlying filters (with $L > k^2$, implying that it is redundant). The dictionary is merely composed of Gaussian and difference of Gaussians (DoG) filters mainly for two reasons. First, as discussed in [33, 34, 35], Gaussians have strong representation ability. Second, difference of Gaussians has been verified to increase the visibility of edges and details [36, 37, 38, 39].

The corresponding elliptical function of Gaussian has the form as

$$G\left(\mathbf{x} - \mathbf{x}'; \mathbf{\Sigma}\right) = \frac{1}{2\pi|\mathbf{\Sigma}|^{\frac{1}{2}}} \exp\{-\frac{1}{2}\left(\mathbf{x} - \mathbf{x}'\right)^T \mathbf{\Sigma}^{-1} \left(\mathbf{x} - \mathbf{x}'\right)\} , \tag{8}$$

where $\mathbf{x}$ and $\mathbf{x}'$ are coordinates of neighboring pixels and central pixel respectively, and the covariance matrix $\mathbf{\Sigma}$ can be decomposed into

$$\mathbf{\Sigma} = \gamma^2 \mathbf{U}_\theta \mathbf{\Lambda} \mathbf{U}_\theta^T ,$$

$$\mathbf{U}_\theta = \begin{bmatrix} \cos\theta & -\sin\theta \\ \sin\theta & \cos\theta \end{bmatrix} , \tag{9}$$

$$\mathbf{\Lambda} = \begin{bmatrix} \sigma_1^2 & 0 \\ 0 & \sigma_2^2 \end{bmatrix} ,$$

where $\gamma, \theta, \sigma_{1/2}$ are scaling, rotation and elongation parameters. The decomposition of Eq. (9) is schematically explained in the left-hand side of Figure 3. Inspired by [5, 40] that train filters based

on local structure statistics, we define the Gaussians with different parameter settings as shown in Figure 3. In general, a Gaussian with a large disparity of $\sigma_1$ and $\sigma_2$ is supposed to preserve edges well. Based on the anisotropic Gaussian kernels in our dictionary, we generate DoG filters as follows,

$$DoG\left(\mathbf{x} - \mathbf{x}'; \boldsymbol{\Sigma}_1, \boldsymbol{\Sigma}_2\right) = G\left(\mathbf{x} - \mathbf{x}'; \boldsymbol{\Sigma}_1\right) - G\left(\mathbf{x} - \mathbf{x}'; \boldsymbol{\Sigma}_2\right), \tag{10}$$

which allows negative values. All 72 filters are normalized to sum total to 1.

## 2.4 Network Architecture

We adopt a lightweight residual network to predict combination coefficients of filters. It consists of multiple local fusion blocks (LFB) [41], a depth-to-space (PixelShuffle) layer and several convolutional layers in the tail. The depth-to-space layer is used to map LR features to HR ones, and the final convolutional layers generate combination coefficients. Besides, weight normalization [26] is employed to accelerate the training process. To evaluate the network scalability, we provide three models according to the number of feature channels ($C$) and local fusion modules ($M$), and name them as LAPAR-A ($C32$-$M4$), LAPAR-B ($C24$-$M3$), and LAPAR-C ($C16$-$M2$) (see also Figure 1).

# 3 Experiments

## 3.1 Datasets and Metrics

In our experiments, the network is trained with DIV2K [42] and Flickr2K image datasets. During the testing stage, based on the peak signal to noise ratio (PSNR) and the structural similarity index (SSIM) [43], multiple standard benchmarks including Set5 [27], Set14 [2], B100 [44], Urban100 [45], Manga109 [46] are used to evaluate the performance of our method. Following previous methods [47, 19, 15], only results of the Y channel from the YCbCr color space are reported in this paper.

## 3.2 Training Details

We use eight NVIDIA GeForce RTX 2080Ti GPUs to train the network with a mini-batch size of 32 for 600K iterations. The optimizer is Adam. We adopt a cosine learning rate strategy with an initial value of $4e - 4$. The input patch size is set to $64 \times 64$. Data augmentation is performed with random cropping, flipping and rotation. The flipping involves vertical or horizontal versions, and the rotation angle is $90°$.

## 3.3 Study of The Filter Dictionary

In this section, we evaluate how the dictionary design affects the final result and also the optimization of the proposed learning strategy.

| Type | Num. | Set5 | B100 |
|---|---|---|---|
| G + DoG | 72 | **38.01** | **32.19** |
| Learned | 72 | 37.98 | **32.19** |
| RAISR [5] | 72 | 37.93 | 32.12 |
| Random | 72 | 37.88 | 32.08 |
| G + DoG | 24 | 37.94 | 32.13 |
| G + DoG | 14 | 37.87 | 32.07 |
| Random | 14 | 37.80 | 32.01 |

Table 1: PSNR(dB) results of different dictionary settings for LAPAR-A on $\times 2$ scale on Set5 and B100.

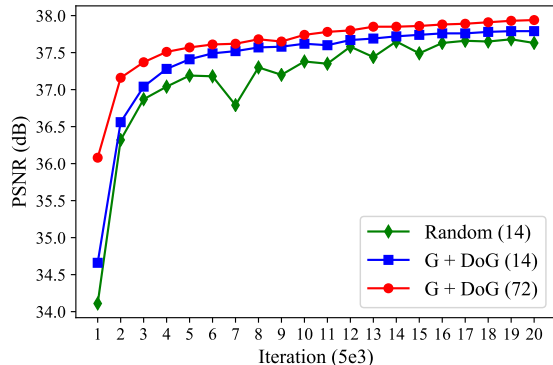

Figure 4: Validation results on Set5 during training.

As aforementioned, we adopt a dictionary composed of 72 Gaussian and DoG filters to capture more structural details. To verify whether the proposed filters are effective or not, we conduct additional experiments by replacing them with random filters or RAISR [5] filters or filters directly learned

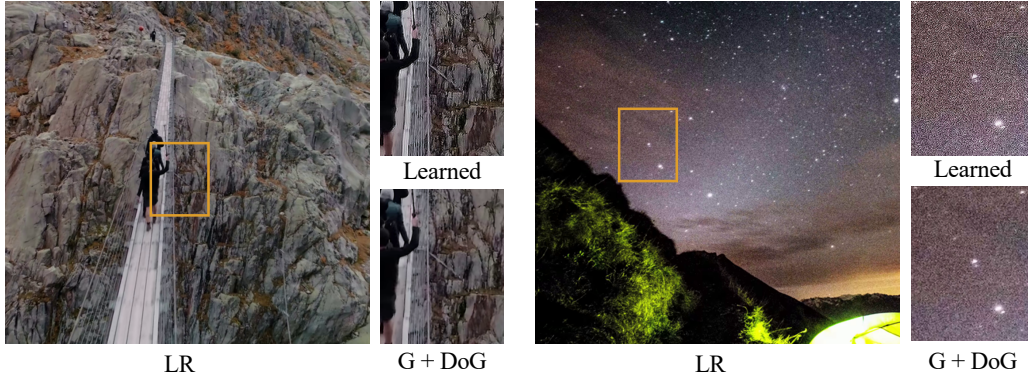

Figure 5: Visualization of the super-resolved images ($\times 2$) generated by learned filters as well as Gaussian and DoG filters. Zoom in for better visual comparison.

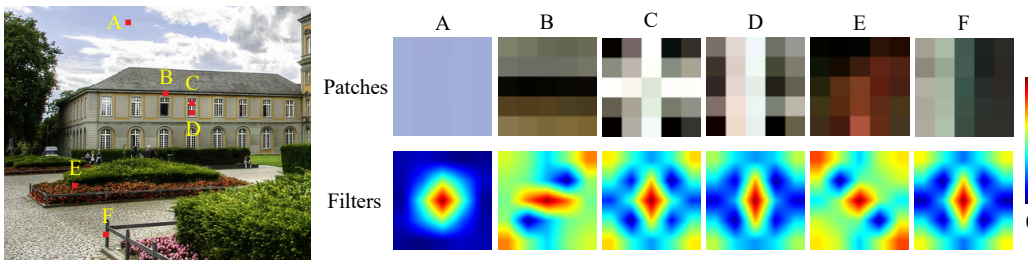

Figure 6: Visualization of the final assembled filters of LAPAR for different pixel locations.

by networks. For fairness, the size of filters learned from RAISR is also set to $5 \times 5$ (contrary to original $11 \times 11$). As reported in Table 1, it is clear that our proposed Gaussian and DoG combination achieves the best result. Although the learned filters obtain comparable results, we find they may sometimes generate artifacts along edges or amplify noise due to overfitting in natural images, as shown in Figure 5. Interestingly, random filters can still achieve competitive performance, which further manifests the feasibility of our proposed learning strategy with the provided dictionary.

We also explore the influence caused by the number of filters on LAPAR-A. Table 1 shows that the dictionary with 72 basic filters yields a better result than the 24 and 14 versions. This study indicates a sufficiently large dictionary with diverse filters is indeed helpful to obtain superior results.

Experiments have also verified that our proposed model enables fast optimization. Figure 4 illustrates the validation results on the Set5 benchmark during training. It is clear that our method reaches a decent level of reconstruction accuracy (e.g., over 37.5dB) after only a small number of iterations. In addition, compared with random filters, the Gaussian and DoG setting is more stable and accurate.

Finally, we visualize some examples of the final assembled filters in Figure 6. By estimating the spatially variant combination coefficients, our model generates adaptive filters to process the structural information differently for each pixel location. In flat areas, such as location A, the kernel tends to be isotropic. For horizontal (B), vertical (D, F) and diagonal (E) edges, one can see that the kernels are well constructed with corresponding orientations. The kernel of the cross-shaped pattern C is similar as D and F, but the weights are more evenly distributed.

## 3.4 Comparison with the State-of-the-Art Methods

To evaluate the performance of our approach, we make a comparison with state-of-the-art lightweight frameworks and show results in Table 2. For the $\times 2$ setting, LAPAR-A outperforms the other methods by a large margin on all benchmark datasets with even fewer parameters and MultiAdds. As the capacity of the network decreases, LAPAR-B and LAPAR-C still obtain competitive results gracefully (see also Figure 1). Even with only 80K parameters, the performance of LAPAR-C is superior to many existing methods. Regarding the $\times 3$ and $\times 4$ setting, our LAPAR-A again stands out as the best.

| Scale | Method | Params | MultiAdds | Set5 | Set14 | B100 | Urban100 | Manga109 |
|---|---|---|---|---|---|---|---|---|
| ×2 | SRCNN [6] | 57K | 53G | 36.66/0.9542 | 32.42/0.9063 | 31.36/0.8879 | 29.50/0.8946 | 35.74/0.9661 |
| | FSRCNN [53] | 12K | 6G | 37.00/0.9558 | 32.63/0.9088 | 31.53/0.8920 | 29.88/0.9020 | 36.67/0.9694 |
| | VDSR [7] | 665K | 613G | 37.53/0.9587 | 33.03/0.9124 | 31.90/0.8960 | 30.76/0.9140 | 37.22/0.9729 |
| | DRCN [9] | 1,774K | 17,974G | 37.63/0.9588 | 33.04/0.9118 | 31.85/0.8942 | 30.75/0.9133 | 37.63/0.9723 |
| | MemNet [54] | 677K | 2,662G | 37.78/0.9597 | 33.28/0.9142 | 32.08/0.8978 | 31.31/0.9195 | - |
| | DRRN [11] | 297K | 6,797G | 37.74/0.9591 | 33.23/0.9136 | 32.05/0.8973 | 31.23/0.9188 | 37.92/0.9760 |
| | LapSRN [10] | 813K | 30G | 37.52/0.9590 | 33.08/0.9130 | 31.80/0.8950 | 30.41/0.9100 | 37.27/0.9740 |
| | SelNet [55] | 974K | 226G | 37.89/0.9598 | 33.61/0.9160 | 32.08/0.8984 | - | - |
| | CARN-M [48] | 412K | 91G | 37.53/0.9583 | 33.26/0.9141 | 31.92/0.8960 | 31.23/0.9193 | - |
| | CARN [48] | 1,592K | 223G | 37.76/0.9590 | 33.52/0.9166 | 32.09/0.8978 | 31.92/0.9256 | - |
| | FALSR-B [56] | 326k | 75G | 37.61/0.9585 | 33.29/0.9143 | 31.97/0.8967 | 31.28/0.9191 | - |
| | FALSR-C [56] | 408k | 94G | 37.66/0.9586 | 33.26/0.9140 | 31.96/0.8965 | 31.24/0.9187 | - |
| | FALSR-A [56] | 1,021K | 235G | 37.82/0.9595 | 33.55/0.9168 | 32.12/0.8987 | 31.93/0.9256 | - |
| | SRMDNF [14] | 1,513K | 348G | 37.79/0.9600 | 33.32/0.9150 | 32.05/0.8980 | 31.33/0.9200 | - |
| | SRFBN-S [47] | 282K | 680G | 37.78/0.9597 | 33.35/0.9156 | 32.00/0.8970 | 31.41/0.9207 | 38.06/0.9757 |
| | **LAPAR-C(Ours)** | **87K** | **35G** | 37.65/0.9593 | 33.20/0.9141 | 31.95/0.8969 | 31.10/0.9178 | 37.75/0.9752 |
| | **LAPAR-B(Ours)** | **250K** | **85G** | 37.87/0.9600 | 33.39/0.9162 | 32.10/0.8987 | 31.62/0.9235 | 38.27/0.9764 |
| | **LAPAR-A(Ours)** | **548K** | **171G** | 38.01/0.9605 | 33.62/0.9183 | 32.19/0.8999 | 32.10/0.9283 | 38.67/0.9772 |
| ×3 | SRCNN [6] | 57K | 53G | 32.75/0.9090 | 29.28/0.8209 | 28.41/0.7863 | 26.24/0.7989 | 30.59/0.9107 |
| | FSRCNN [53] | 12K | 5G | 33.16/0.9140 | 29.43/0.8242 | 28.53/0.7910 | 26.43/0.8080 | 30.98/0.9212 |
| | VDSR [7] | 665K | 613G | 33.66/0.9213 | 29.77/0.8314 | 28.82/0.7976 | 27.14/0.8279 | 32.01/0.9310 |
| | DRCN [9] | 1,774K | 17,974G | 33.82/0.9226 | 29.76/0.8311 | 28.80/0.7963 | 27.15/0.8276 | 32.31/0.9328 |
| | MemNet [54] | 677K | 2,662G | 34.09/0.9248 | 30.00/0.8350 | 28.96/0.8001 | 27.56/0.8376 | - |
| | DRRN [11] | 297K | 6,797G | 34.03/0.9244 | 29.96/0.8349 | 28.95/0.8004 | 27.53/0.8378 | 32.74/0.9390 |
| | SelNet [55] | 1,159K | 120G | 34.27/0.9257 | 30.30/0.8399 | 28.97/0.8025 | - | - |
| | CARN-M [48] | 412K | 46G | 33.99/0.9236 | 30.08/0.8367 | 28.91/0.8000 | 27.55/0.8385 | - |
| | CARN [48] | 1,592K | 119G | 34.29/0.9255 | 30.29/0.8407 | 29.06/0.8034 | 28.06/0.8493 | - |
| | SRMDNF [14] | 1,530K | 156G | 34.12/0.9250 | 30.04/0.8370 | 28.97/0.8030 | 27.57/0.8400 | - |
| | SRFBN-S [47] | 376K | 832G | 34.20/0.9255 | 30.10/0.8372 | 28.96/0.8010 | 27.66/0.8415 | 33.02/0.9404 |
| | **LAPAR-C(Ours)** | **99K** | **28G** | 33.91/0.9235 | 30.02/0.8358 | 28.90/0.7998 | 27.42/0.8355 | 32.54/0.9373 |
| | **LAPAR-B(Ours)** | **276K** | **61G** | 34.20/0.9256 | 30.17/0.8387 | 29.03/0.8032 | 27.85/0.8459 | 33.15/0.9417 |
| | **LAPAR-A(Ours)** | **594K** | **114G** | 34.36/0.9267 | 30.34/0.8421 | 29.11/0.8054 | 28.15/0.8523 | 33.51/0.9441 |
| ×4 | SRCNN [6] | 57K | 53G | 30.48/0.8628 | 27.49/0.7503 | 26.90/0.7101 | 24.52/0.7221 | 27.66/0.8505 |
| | FSRCNN [53] | 12K | 5G | 30.71/0.8657 | 27.59/0.7535 | 26.98/0.7150 | 24.62/0.7280 | 27.90/0.8517 |
| | VDSR [7] | 665K | 613G | 31.35/0.8838 | 28.01/0.7674 | 27.29/0.7251 | 25.18/0.7524 | 28.83/0.8809 |
| | DRCN [9] | 1,774K | 17,974G | 31.53/0.8854 | 28.02/0.7670 | 27.23/0.7233 | 25.14/0.7510 | 28.98/0.8816 |
| | MemNet [54] | 677K | 2,662G | 31.74/0.8893 | 28.26/0.7723 | 27.40/0.7281 | 25.50/0.7630 | - |
| | DRRN [11] | 297K | 6,797G | 31.68/0.8888 | 28.21/0.7720 | 27.38/0.7284 | 25.44/0.7638 | 29.46/0.8960 |
| | LapSRN [10] | 813K | 149G | 31.54/0.8850 | 28.19/0.7720 | 27.32/0.7280 | 25.21/0.7560 | 29.09/0.8845 |
| | SelNet [55] | 1,417K | 83G | 32.00/0.8931 | 28.49/0.7783 | 27.44/0.7325 | - | - |
| | SRDenseNet [57] | 2,015K | 390G | 32.02/0.8934 | 28.50/0.7782 | 27.53/0.7337 | 26.05/0.7819 | - |
| | CARN-M [48] | 412K | 33G | 31.92/0.8903 | 28.42/0.7762 | 27.44/0.7304 | 25.62/0.7694 | - |
| | CARN [48] | 1,592K | 91G | 32.13/0.8937 | 28.60/0.7806 | 27.58/0.7349 | 26.07/0.7837 | - |
| | SRMDNF [14] | 1,555K | 89G | 31.96/0.8930 | 28.35/0.7770 | 27.49/0.7340 | 25.68/0.7730 | - |
| | SRFBN-S [47] | 483K | 1,037G | 31.98/0.8923 | 28.45/0.7779 | 27.44/0.7313 | 25.71/0.7719 | 29.91/0.9008 |
| | **LAPAR-C(Ours)** | **115K** | **25G** | 31.72/0.8884 | 28.31/0.7740 | 27.40/0.7292 | 25.49/0.7651 | 29.50/0.8951 |
| | **LAPAR-B(Ours)** | **313K** | **53G** | 31.94/0.8917 | 28.46/0.7784 | 27.52/0.7335 | 25.85/0.7772 | 30.03/0.9025 |
| | **LAPAR-A(Ours)** | **659K** | **94G** | 32.15/0.8944 | 28.61/0.7818 | 27.61/0.7366 | 26.14/0.7871 | 30.42/0.9074 |

Table 2: Comparisons on multiple benchmark datasets for lightweight networks. The MultiAdds is calculated corresponding to a $1280 \times 720$ HR image. **Bold**/red/blue: **our**/best/second best results.

Although CARN [48] obtains comparable PSNR results, our method requires fewer parameters and performs better on SSIM. It further verifies that our method can lead to higher structural similarity. Meanwhile, compared with large EDSR [49], RCAN [50] and ESRGAN [51] and ProSR [52] that have parameters/PSNR as 43M/27.71dB, 16M/27.77dB, 17M/27.76dB and 16M/27.79dB under $\times 4$ setting on the B100 dataset, our LAPAR-A model (0.66M/27.61dB) is $\mathbf{25\times - 65\times}$ smaller, yet achieving quite similar super-resolution results. All these results justify the effectiveness of our method.

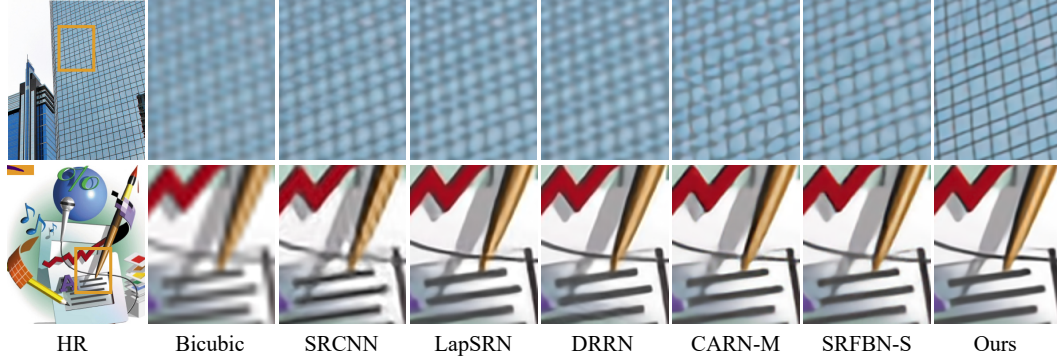

| HR | Bicubic | SRCNN | LapSRN | DRRN | CARN-M | SRFBN-S | Ours |

Figure 7: Image super-resolution examples on $\times 4$ scale of Urban100 and Set14.

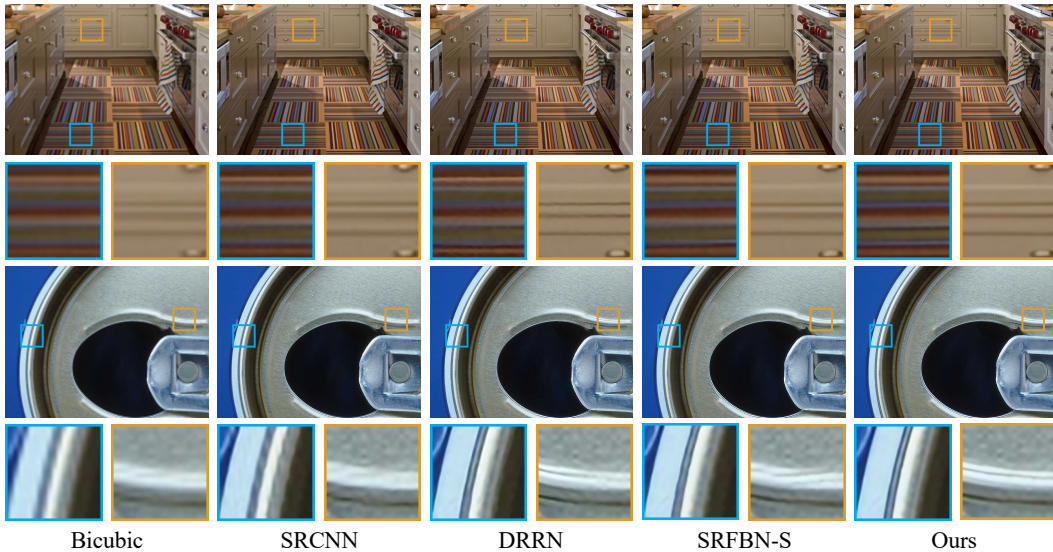

| Bicubic | SRCNN | DRRN | SRFBN-S | Ours |

Figure 8: Image super-resolution examples on $\times 4$ scale of general cases in the wild.

Our method has fast inference speed. To obtain a $1280 \times 720$ output in the $\times 4$ setting, LAPAR-A, LAPAR-B and LAPAR-C cost 37.3ms, 29.1ms and 22.2ms on a single NVIDIA 2080Ti GPU.

Examples are visualized in Figure 7 and Figure 8. Compared with other methods, LAPAR can generate results with better visual effects. The structures and details are clearly better recovered.

### 3.5 Image Denoising and JPEG Image Deblocking

In this part, we make further exploration of image denoising and deblocking based on LAPAR.

**Image Denoising.** Following [40, 58], the noisy images are synthesized. To handle noise with different levels, we train the model with a broad noise standard deviation range (i.e., $\sigma_{noise} \in [0, 55]$). Besides, the upsampling layer of $LaparNet$ in Figure 2 is removed. The predicted filters directly work on the noisy input. We stick to the original filter dictionary and optimize the model parameters in the same manner as the presented SISR task. The denoised results in Figure 9 show that LAPAR removes the blind noise effectively. Compared with other methods [59, 58], LAPAR achieves impressive performance on restoring the original color and structure of the test images.

**JPEG Image Deblocking.** Besides denoising, LAPAR is also good at JPEG image deblocking. During training, the inputs to the network are JPEG-compressed images with varying quality of $20 \sim 50$. As the testing cases shown in Figure 10, our method removes the JPEG artifacts successfully and obtains superior results compared with DnCNN [58] that is an effective deblocking approach.

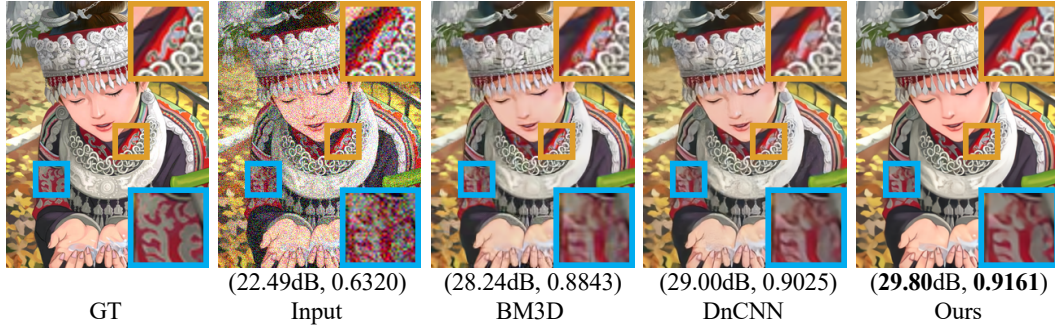

| | (22.49dB, 0.6320) | (28.24dB, 0.8843) | (29.00dB, 0.9025) | (**29.80**dB, **0.9161**) |
| GT | Input | BM3D | DnCNN | Ours |

Figure 9: Denoising examples. The values beneath images represent the PSNR(dB) and SSIM. The standard deviation of noise is set to 35.

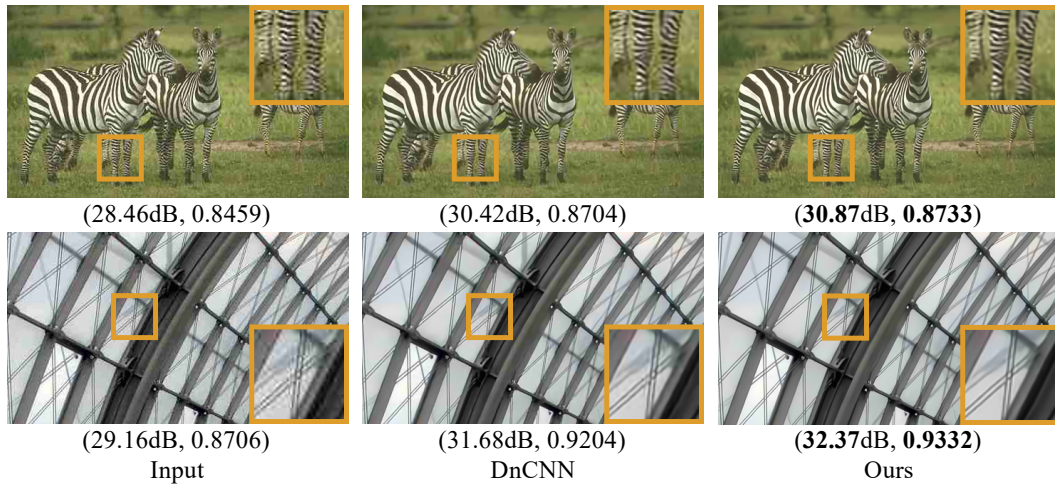

| (28.46dB, 0.8459) | (30.42dB, 0.8704) | (**30.87**dB, **0.8733**) |
| (29.16dB, 0.8706) | (31.68dB, 0.9204) | (**32.37**dB, **0.9332**) |
| Input | DnCNN | Ours |

Figure 10: JPEG image deblocking examples. The values beneath images represent the PSNR(dB) and SSIM. The JPEG quality is set to 20.

## 4  Conclusion

We have presented a linearly-assembled pixel-adaptive regression network (LAPAR) for image super-resolution. Extensive experiments have demonstrated the effectiveness of the proposed learning strategy. Among lightweight methods, LAPAR achieves state-of-the-art results on multiple benchmarks. Besides, we also show that LAPAR is easily extended to other low-level restoration tasks e.g., denoising and JPEG deblocking, and obtains decent result quality. In future work, we plan to investigate other compact representations of the filter dictionary and joint multi-task optimization.

## Broader Impact

This paper aims to promote academic development. In the past decades, image super-resolution, denoising and deblocking techniques have marked new milestones and also been widely used in industry. As far as we know, they have no negative impact on the ethical and societal aspects.

## Acknowledgments and Disclosure of Funding

GPUs supported by SmartMore Technology.

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
