[Supplementary Material]

# Supplementary Material of LAPAR

**Wenbo Li[1]*   Kun Zhou[2]*   Lu Qi[1]   Nianjuan Jiang[2]   Jiangbo Lu[2]   Jiaya Jia[1,2]**

[1]The Chinese University of Hong Kong    [2]Smartmore Technology

{wenboli,luqi,leojia}@cse.cuhk.edu.hk
{kun.zhou,nianjuan.jiang,jiangbo}@smartmore.com

## A. Additional Examples and Results of Image Super-Resolution

Here we show more visual examples on the Urban100 dataset in Figure 1. For the first example, it is clear that our LAPAR recovers more accurate structures while other methods [1, 2, 3, 4, 5] fail. For the second one, although other methods produce building transoms and mullions, our results are obviously sharper and straighter.

Figure 1: Image super-resolution examples on ×2 (top part) and ×4 (bottom part) scale of Urban100.

| Method | Scale | Params | MultiAdds | Set5 | Set14 | B100 | Urban100 | Manga109 |
|--------|-------|--------|-----------|------|-------|------|----------|----------|
| | ×2 | 0.548M | 171G | 37.95/38.01 | 33.58/33.62 | 32.17/32.19 | 32.01/32.10 | 38.41/38.67 |
| LAPAR-A | ×3 | 0.594M | 114G | 34.31/34.36 | 30.30/30.34 | 29.06/29.11 | 28.10/28.15 | 33.31/33.51 |
| | ×4 | 0.659M | 94G | 32.10/32.15 | 28.53/28.61 | 27.56/27.61 | 26.01/26.14 | 30.22/30.42 |

Table 1: PSNR(dB) results of LAPAR-A. Red/blue: trained on DIV2K/DIV2K+Flickr2K.

---

As shown in Table 1, we also show the results of our LAPAR trained only on DIV2K. LAPAR-A *still* achieves SOTA performance among lightweight SISR methods. Besides, we compare the results of RAISR [6] and LAPAR-A in Table 2, it is clear that our method outperforms RAISR [6] by a large margin.

| Method | Scale | Set5 | Set14 |
|---|---|---|---|
| RAISR [6] | $\times 2$ | 36.15/0.951 | 32.13/0.902 |
| | $\times 3$ | 32.21/0.901 | 28.86/0.812 |
| | $\times 4$ | 29.84/0.848 | 27.00/0.738 |
| LAPAR-A | $\times 2$ | 38.01/0.961 | 33.62/0.918 |
| | $\times 3$ | 34.36/0.927 | 30.34/0.842 |
| | $\times 4$ | 32.15/0.894 | 28.61/0.782 |

Table 2: Comparison of RAISR [6] and LAPAR-A. The values represent PSNR(dB)/SSIM.

## B. Additional Examples of Image Denoising

As shown in Figure 2, more denoised examples of Set14 dataset are visualized. Compared with other methods [7, 8], for the first example, our LAPAR restores the original white background color nicely. At the same time, the details of all the pictures are better preserved in our results.

| (23.20dB, 0.4573) | (28.22dB, 0.9368) | (28.45dB, 0.9696) | (**36.94**dB, **0.9865**) |
| (22.72dB, 0.4595) | (30.18dB, 0.8510) | (30.69dB, 0.8719) | (**31.75**dB, **0.8867**) |
| (22.61dB, 0.3289) | (32.70dB, 0.8602) | (32.28dB, 0.8649) | (**34.02**dB, **0.8722**) |
| Input | BM3D | DnCNN | Ours |

Figure 2: Image denoising examples of Set14. The values beneath images represent the PSNR(dB) and SSIM. The standard deviation of noise is set to 35.

## C. Additional Examples of Image Deblocking

As the testing cases illustrated in Figure 3, our LAPAR successfully removes the JPEG compression artifacts and achieves superior results compared with DnCNN [8].

(30.44dB, 0.8822)        (32.62dB, 0.9156)        (**33.36**dB, **0.9243**)

(33.68dB, 0.9156)        (35.80dB, 0.9429)        (**36.39**dB, **0.9457**)

(33.53dB, 0.9304)        (36.65dB, 0.9612)        (**36.92**dB, **0.9631**)

Input             DnCNN             Ours

Figure 3: Image deblocking examples of Set14. The values beneath images represent the PSNR(dB) and SSIM. The JPEG quality is set to 20.