[Reviews · NeurIPS 2020]

Review 1

Summary and Contributions: The authors propose a method for single image super resolution that can be described in three steps: The first step learns to predict filter coefficients which are used in a second step to assemble filters based on a (manually) predefined dictionary of filters. In the third and last step, these filters are applied to a bicubic upscale of the input to refine it and to form the final high res output.

Strengths: By splitting the upscaling tasks in these steps, the first part of the network responsible for estimating filter coefficients can remain relatively small in terms of parameters compared to methods that directly predict high resolution outputs without filter kernels. The results seem to be state of the art and also the number of multiply add operations performed during the upscale is smaller when compared to existing methods. The algorithm seems to work best on content that is fairly flat but has a few detailed lines / text / or fine structures.

Weaknesses: Overall, the idea of predefining a dictionary of filters and to then learn to predict corresponding filter coefficients is interesting and does seem to work quite well. However, technically this idea may not be complex enough and therefore the contribution within this paper may not be large enough for NeurIPS. Concerning upscaling results, it would be interesting to see more upscales with fine grained texture and it would have been interesting to see whether kernels also allow for good performance when used in a GAN context.

Correctness: The claims seem correct.

Clarity: The paper is relatively well written.

Relation to Prior Work: Relation to prior work seems to be discussed sufficiently well. In single image super resolution the body of existing works is extensive. Additional comparisons to e.g. EDSR, RCAN, ESRGAN, ProSR should be included (in terms of objective numbers and visually).

Reproducibility: Yes

Additional Feedback: I would be interested to know whether the proposed approach also lends itself to GAN training or whether the predefined kernels inherently constrain the output result to make more complex hallucination of detail difficult. How does the method perform in the real world when various content is used as input that might have undergone different processing beforehand (and was not downscaled)?


Review 2

Summary and Contributions: This paper proposed a linearly-assembled pixel-adaptive regression network (LAPAR) for SISR. Experiment results show that LAPAR get strong performance with fewer parameters and less computation cost for SISR.

Strengths: It is great to incorporate the concept of dictionary learning into neural network design, for the task of image restoration.

Weaknesses: The dictionary used in reconstructing HR images is hand-crafted. Why can the filters in the dictionary not be learned as kernels in neural network and enjoy the benefit of end-to-end learning as many pure deep learning-based SISR method? In the experiment, when comparing with SOTA SISA methods, only x2 and x4 results are shown while x3 results are missing. The authors are recommended to provide x3 results as well. In addition, FALSR-C and FALSR-A in Table 2 used only DIV2K as the training set, while the training set of the proposed method are both DIV2K and Flickr2K, and thus the comparison here is not fair. The authors are recommended to report the result of the proposed method trained only on DIV2K. There are not enough experiment results for the task of image denoising and JPEG de-blocking. The authors should at least report the quantitative results over benchmark datasets to show the performance, rather than displaying only a few image examples. Without such evidence, the claim about the proposed method for these two tasks is not well-founded.

Correctness: Yes

Clarity: Yes

Relation to Prior Work: Yes

Reproducibility: Yes

Additional Feedback:


Review 3

Summary and Contributions: This paper aims to develop a state-of-the-art solution to single image super-resolution, through a combination of learned linear combination coefficients and preselected filters to determine the relationship between the high-resolution image and the bicubic interpolation low-resolution counterpart. The unique lightweight architecture of the LAPAR network combined with the addition of the local filters are new contributions to the field of super-resolution.

Strengths: The work presented by the authors is significant in creating fast and lightweight super resolution images, as it can be potentially useful in commercial applications.

Weaknesses: There are some concerns: 1. In line 82, authors should provide more explanations why they assumed linear constrains. How does it compare with non-linear combination in terms of performance and optimization speed. 2. How to prove the pre-defined dictionary is over-complete? How to compare the hand-crafted filters with learned filters? Experiments on Set5 is limited in data size and generalization ability. 3. How does the cheap upsampling method (bicubic in the paper) influence the result? What is the limitations of upscaling factor, say will it fail if the factor is 8? 4. More comparisons and results from RAISR should be presented. 5. Experiments on image denoising and deblocking is very limited, lacking quantitative comparisons on benchmarks and intuitive explanation of this generalization.

Correctness: I expect more explanations on linear constrains and justifications of the over-complete filter bank size.

Clarity: The authors present a well-written paper demonstrating a general overview of the advances that they have claimed in the field of super-resolution. The authors inclusion of the three versions (LAPAR-A, LAPAR-B, and LAPAR-C) confuse the reader in the results, as it is not clear which visual result is included in the side-by-side comparisons, and possibly could be cherry-picked for visual performance.

Relation to Prior Work: The work presents a shortcut towards super resolution that cuts down on model complexity and makes new advances in resolution accuracy. This discussion is evident in the comparisons of architecture towards previous work, and the advances in decreasing the number of MultiAdds and parameters while increasing resolution performance separate itself from other works in the same field.

Reproducibility: Yes

Additional Feedback: I do think a light-weight SR network is interesting and a smaller dictionary may be more efficient while being deployed into mobile devices. However, this paper is still not ready for publication. According to other reviewers, some of state-of-the-arts are not compared, even though they have more parameters. I also have concerns on their linearity assumptions and the ablation study. So I changed my rating from 6 to 5, and hopefully the authors could revise the paper and submit it again.


Review 4

Summary and Contributions: The authors proposed a linearly-assembled pixel-adaptive regression network and applied it to image super-resolution, denoising, and compression artifact reduction.

Strengths: The authors proposed a light-weight network and compared with several previous related works. Experiments about super-resolution, denoising, and compression artifact reduction.

Weaknesses: The comparisons with previous methods are not fair. The authors used more training data: DIV2K+Flickr2K. Most compared methods used much less training data. For image SR, the authors didn’t compare with other state-of-the-art methods, like EDSR, RCAN. For image denoising and JPEG deblocking, the authors didn’t compare with more recent methods. Even for DnCNN, the comparison is not fair, because the authors used much more high-quality training data.

Correctness: The empirical methodology is correct.

Clarity: The paper writing is easy to understand and follow.

Relation to Prior Work: The authors didn’t show sufficient discussions or experiments about why their method is better than most previous ones, like EDSR and RCAN. The current performance gains over most light-weight networks come from the usage of much larger training data, input size, and batch size.

Reproducibility: Yes

Additional Feedback:

[Author Response · NeurIPS 2020]

# Rebuttal of LAPAR (Paper ID: 653)

We highly appreciate all of your constructive comments as well as your recognition of this work's interesting perspective
of rethinking and designing a lightweight and practical SISR method. We'll first answer some questions in common.

**1. Comparison with EDSR, RCAN, ESRGAN and ProSR.** As motivated in the paper, we aim to design a lightweight
and real-time SISR method by exploring a novel idea of linear coefficient regression over a dictionary of filter bases.
Thus, the models we compared are also small models ($< 1$ M params.). By contrast, EDSR, RCAN and ESRGAN and
ProSR have params./PSNR as 43M/27.71dB, 16M/27.77dB, 17M/27.76dB and 16M/27.79dB under $\times 4$ setting on B100
dataset, which are $\mathbf{20 - 400\times}$ larger than LAPAR-A (**0.66M/27.56dB**) with only 0.2dB gain. We'll add comparisons.

**2. Results trained only on DIV2K.** We followed exactly SRFBN, USRNet, DRN, UDVD to use both DIV2K and
Flickr2K datasets. Here we also show the results of our LAPAR trained **only on DIV2K**. LAPAR-A *still* achieves SOTA
performance among lightweight SISR methods. <span style="color:red">Red</span>/<span style="color:blue">blue</span>: PSNR(dB) results trained on <span style="color:red">DIV2K</span>/<span style="color:blue">DIV2K+Flickr2K</span>.

| Method | Scale | Params | MultiAdds | Set5 | Set14 | B100 | Urban100 | Manga109 |
|--------|-------|--------|-----------|------|-------|------|----------|----------|
| LAPAR-A | $\times 2$ | 0.548M | 171G | 37.95/38.01 | 33.58/33.62 | 32.17/32.19 | 32.01/32.10 | 38.41/38.67 |
| | $\times 4$ | 0.659M | 94G | 32.10/32.15 | 28.53/28.61 | 27.56/27.61 | 26.01/26.14 | 30.22/30.42 |

**3. There are not enough results for the task of image denoising and JPEG de-blocking.** Like RAISR [5] and
BLADE [39], we briefly studied the usefulness of LAPAR in image denoising and JEPG deblocking tasks, with the
purpose to suggest the extensible applications of the proposed framework. Actually, it is *not* our intention to claim the
SOTA performance of LAPAR. In view of the limited space, we will shrink this part to focus more on SISR discussions.

**4. Hand-crafted filters v.s. learned filters.** In our experiments, we found the learned filters may sometimes generate
artifacts along edges due to overfitting. By contrast, the predefined meaningful Gaussian and DoG filters perform
stably, also allowing for flexible control of the dictionary size and the consequent computational complexity in practice.
Nevertheless, as also admitted in paper conclusions, it's indeed an interesting direction to study coupled optimization.

[*Reviewer* #1]
**1. Whether the proposed approach also lends itself to GAN training.** Thanks. LAPAR is a plug and play method
and can lend itself easily to GAN training by adopting a discriminator. However, most GAN-based methods require
generators with large capacity, which deviates from our current design objective. We'll explore this idea in the future.

**2. Whether the predefined kernels inherently make it difficult to make more complex hallucination.** As shown
in L99-102, previous work demonstrated the strong representation ability of Gaussians and DoGs. Also, aided by the
estimated combination coefficients, LAPAR can aggregate neighboring pixels to reconstruct arbitrary target values,
hence allowing it to make complex hallucination in both fine-grained and flat areas. We'll elaborate more on this point.

**3. Performance in the real world.** As shown in Figure 7, our LAPAR performs well on the real-world images. Thanks,
we will provide more visual examples with various processing beforehand in the supplementary material.

[*Reviewer* #2]
**1. The $\times 3$ results are missing.** We'll add the results of $\times 3$ setting. We saw similar conclusions as $\times 2/\times 4$ settings.

[*Reviewer* #3]
**1. Linear v.s. non-linear combination in terms of performance and optimization speed.** For simplicity and speed,
we adopt the linear combination, which facilitates the optimization as verified in SMPL [31] and LSM [Tang *et al.* ,
CVPR 2020]. Thanks for this nice suggestion. Exploring a non-linear constraint will be an interesting future direction.

**2. The predefined dictionary is over-complete or not.** As stated in L98-99, our dictionary is redundant but not
intended to be over-complete (a good future study topic), just like RAISR [5]. However, RAISR only designates fixed
filters, we propose to linearly assemble filter bases to further improve the representation ability of dictionary. Actually,
based on the linear combination weights regressed for every given pixel, LAPAR can reconstruct any target value for it.

**3. Experiments on Set5 is limited in data size and generalization ability.** We also obtained the same conclusions
on B100, Urban100 and Manga109 datasets. Thanks for useful suggestions. We will add these results into the paper.

**4. How does the cheap upsampling method (bicubic in the paper) influence the result? What is the limitation of
upscaling factor?** In our experiments, it has little impact on the recovery result by replacing the bicubic with bilinear.
Under the $\times 8$ setting, in our offline experiments, LAPAR can still work well. Thank you, and we will add the results.

**5.** LAPAR-A is used for *all* visual comparisons. We'll add more comparisons with RAISR (2dB lower than LAPAR-A).

[*Reviewer* #4]
**1. Input size and batch size settings.** Thanks. Our settings just followed previous work e.g. CARN, DRN, UDVD,
USRNet, SAN. Even after reducing the batch size from 32 to 16, LAPAR still yields approximate results ($\pm 0.02$dB).

[Meta-Review · NeurIPS 2020]

This submission proposes to do single image super-resolution using a network which produces coefficients for a fixed bank of Gaussian/DoG filters. The super-resolution results produce nearly SotA super-resolution PSNR while the proposed approach is 1-2 orders of magnitude more efficient than SotA. Strengths: - Novel domain-specific architecture for super-resolution. Reviewers liked the idea of incorporating a filter bank dictionary. - The proposed approach achieves nearly SotA super-resolution results in terms of PSNR. - The proposed "LaparNet" is 20-400x more efficient (in terms of MultAdds) than the SotA approaches. - Qualitative results look subjectively good. - The proposed approach could be of interest to researchers in other domains involving "upsampling," e.g. GAN-based image generation. Weaknesses: - (W1) Comparisons vs. SotA missing. Slightly worse than these SotA approaches in PSNR. - (W2) The filter bank is hand-crafted, not learnt. - (W3) Comparisons in certain standard regimes missing from the original submission: 3x super-resolution results, pretraining only on DIV2K rather than both DIV2K & Flickr2K. - (W4) Not a complex enough idea. While all of the reviewers felt that these weaknesses put the submission below the acceptance threshold, metareviewers felt that the authors' response adequately addressed each of these concerns. (W1) This must be addressed in the camera-ready update as the authors have promised in the rebuttal. Please add comparisons with the SotA approaches (EDSR, RCAN, ESRGAN, ProSR) in terms of PSNR, efficiency (MultAdds), and parameter count. From the authors' response, each of these networks are orders of magnitude larger than the proposed approach while only performing ~1% better than the proposed approach in terms of PSNR (e.g. 27.79dB for ProSR vs. 27.56dB for LAPAR). (W2) While it may be surprising in the deep learning era that hand-crafted filters outperform learnt ones, it's not necessarily a weakness (let alone one that should be considered a "deal-breaker" for an otherwise worthy submission). Given strong results, learning less of the network is ultimately a simplification rather than a real problem. The chosen filter bank also seems to be well-motivated by prior work in this space. However, it would be useful if the camera-ready version would add quantitative comparisons with a learnt filter bank to reinforce the authors' observation in the rebuttal that learning this filter bank results in more overfitting. (W3) The rebuttal addressed this sufficiently by including inlined additional results for training only on DIV2K (and also observing that prior work has operated in the same protocol of pretraining on DIV2K+Flickr2K, so it's clear that this was not done with any intent to unfairly mislead), and the promise to include results for 3x superresolution in the final results. These results should be included in the camera-ready version as promised. (W4) Complexity should only be encouraged where it's really needed. In this domain a simple method seems to achieve strong results. Given that the method achieves strong results nearly on par with SotA despite using a much smaller and more efficient network, and the weaknesses pointed out by reviewers have been addressed sufficiently, the paper is above the acceptance threshold, conditioned on the addition of the additional results and discussion that the authors promised in their rebuttal.